# Aryl Hydrocarbon Receptor Repressor Is Hypomethylated in Psoriasis and Promotes Psoriasis-like Inflammation in HaCaT Cells

**DOI:** 10.3390/ijms222312715

**Published:** 2021-11-24

**Authors:** Ji-Young Um, Bo-Young Chung, Han-Bi Kim, Jin-Cheol Kim, Chun-Wook Park, Hye-One Kim

**Affiliations:** Department of Dermatology, College of Medicine, Hallym University, Kangnam Sacred Heart Hospital, Seoul 150-950, Korea; ujy0402@hanmail.net (J.-Y.U.); victoryby@naver.com (B.-Y.C.); khmamy1029@naver.com (H.-B.K.); aiekfne@naver.com (J.-C.K.); dermap@hanmail.net (C.-W.P.)

**Keywords:** aryl hydrocarbon receptor repressor, CYP1A1, psoriasis, methylation, TCDD, 2,3,7,8-tetrachlorodibenzo-p-dioxin, aryl hydrocarbon receptor

## Abstract

It is known that DNA hypomethylation of aryl hydrocarbon receptor repressor (AhRR), one of the epigenetic markers of environmental pollutants, causes skin diseases. However, the function and mechanisms are still unknown. We aimed to determine whether AhRR is hypomethylated in PBMC of psoriasis patients, as well as to examine the expression of psoriasis-related inflammatory cytokines and antimicrobial peptides after 2,3,7,8-tetrachlorodibenzo-p-dioxin (TCDD) treatment in HaCaT cells overexpressing or silencing AhRR. AhRR was determined by qPCR, Western blot, immunohistochemistry, and immunocytochemistry in skin tissue and HaCaT cells. DNA methylation of AhRR was performed by Infinium Human Methylation450 BeadChip in PBMC of psoriasis patients and methylation-specific PCR (MSP) in HaCaT cells. NF-κB pp50 translocation and activity were performed by immunocytochemistry and luciferase reporter assay, respectively. We verified *AhRR* gene expression in the epidermis from psoriasis patients and healthy controls. *AhRR* hypomethylation in PBMC of psoriasis patients and pAhRR-HaCaT cells was confirmed. The expression level of AhRR was increased in both TCDD-treated HaCaT cells and pAhRR-HaCaT cells. NF-κB pp50 translocation and activity increased with TCDD. Our results showed that *AhRR* was hypomethylated and overexpressed in the lesional skin of patients with psoriasis, thereby increasing AhRR gene expression and regulating pro-inflammatory cytokines through the NF-κB signaling pathway in TCDD-treated HaCaT cells.

## 1. Introduction

Psoriasis is generally thought to be an inflammatory disease of the skin, one that is closely related to environmental pollutants [1,2]. For example, very common environmental factors such as exposure to air pollutants and tobacco smoke are known to play an important role in psoriasis [3]. There is growing evidence that environmental pollution such as this contributes to these skin conditions [4].

Biological responses to many environmental pollutants include direct interaction of xenobiotic receptors with aryl hydrocarbon receptors (AhRs) that bind to several exogenous ligands such as 2,3,7,8-tetrachlorodibenzo-p-dioxin (TCDD) [5]. It is the most recognizable and potent member of the halogenated aromatic hydrocarbon family, which consists of a variety of environmental pollutants [6,7]. With this, TCDD exposure has adverse effects on development and inflammation [4]. The expression of AhR was increased in the lesional skin of psoriasis in our previous study [8]. It was also confirmed in our previous study that it is associated with T cell and AhR signaling by TCDD [9]. Results showed that AhR signaling mediates immunosuppression, as well as crosstalk between AhR and inflammatory responses [9,10,11,12]. In addition, TCDD induces the expression of pro-inflammatory cytokines such as IL-1β, IL-6, IL-8, or TNFα in the lesion skin of psoriasis patients [8,13].

AhRR acts as a specific inhibitor of AhR activity through competition with AhR to form heterodimers with aryl hydrocarbon receptor nuclear translocator (ARNT), which hinders the binding and transactivation of AhR/ARNT complexes via dioxin response elements (DREs) [14]. AhRR plays a role in regulating AhR signaling in key cellular processes, such as cell growth and inflammation [14]. This process is complex and subject to cellular and environmental pollutants [14].

Recent research has reported a link between the epigenetics of AhRR and environmental factors [15]. Epigenetic modifications of the regulatory domain of AhRR were found to be related to tobacco smoke exposure in human studies and the development of several different types of cancer [16]. Recently, the effects of epigenetic alterations on the pathogenesis of psoriasis have been studied [17]. It is also known that changes in DNA methylation affect inflammation [18]. DNA methylation by various environmental exposures, such as air pollution, temperature, radiation, and tobacco smoke, can be associated with altered gene expression [15,19]. This is essential for normal development and is associated with various physiological and pathological processes, including inhibition of implantable elements, aging, and carcinogenesis. For example, AhRR hypomethylation is affected by tobacco smoking during the human fetal period [18]. In addition, recent study has shown that hypomethylation caused by tobacco smoke affects chronic inflammatory diseases in lung organs [20]. AhRR methylation isolated from human blood has demonstrated hypomethylation following long-term smoking exposure [21,22]. It is not yet clear whether AhRR methylation affects inflammatory skin diseases. The aim of this study was to determine whether methylation and expression of AhRR plays a role in psoriasis, and to investigate the mechanism involved.

## 2. Results

### 2.1. Messenger RNA and Protein Expression of AhRR in Human Skin Tissue

We performed immunohistochemistry to investigate the localization of AhRR expression in five patients with psoriasis and five healthy controls. *AhRR* gene expression was significantly increased in the whole layers of the epidermis of psoriasis more than in the healthy skin group (Figure 1A). Simultaneously, epidermal *AhRR* and *CYP1A1* mRNA expression in psoriasis skin tissue was significantly increased compared to healthy skin tissue (Figure 1B). Several environmental carcinogens such as TCDD are known to affect CYP1A1 expression. Therefore, it was confirmed as a marker as to whether AhR was activated by TCDD. These results suggest that AhRR expression is increased in psoriatic skin patients.

### 2.2. Hypomethylation of AhRR in PBMC of Psoriasis Patients and HaCaT Cells

DNA methylation from peripheral blood mononuclear cells (PBMC) of psoriasis patients and PBMC of healthy volunteers exhibited hypomethylation in several loci of the AhRR gene in patients with psoriasis (Figure 2A and Appendix A). We considered cigarette smoke as one of the common environmental factors and divided the patients into normal and psoriasis patients with or without smoking. The results showed that hypomethylation of the *AhRR* gene was dominant in current smoking patients with psoriasis in the body loci of the gene. Appendix A adds the number of samples with and without smoking in normal and psoriasis patients. In addition, AhRR methylation of HaCaT cells was confirmed by methylation-specific PCR (Figure 2B). AhRR overexpression (pAhRR) and TCDD-treated HaCaT cells showed significant hypomethylation of the AhRR gene in comparison with non-treated cells. These results suggest that induced AhRR hypomethylation in the PBMC of smoking patients with psoriasis. These results showed that AhRR hypomethylation was induced in both TCDD-treated HaCaT cells and AhRR-ocerexpressing HaCaT cells. Statistical significance of differences between groups was assessed by one-way analysis of variance (ANOVA) for factorial comparisons and by Tukey’s multiple comparison tests for multiple comparisons. Although the effect of AhRR on cigarette smoke-induced DNA methylation needs to be further examined, the results of TCDD promote AhRR hypomethylation and gene expression, as well as suggesting an association between AhRR and psoriasis in HaCaT cells.

### 2.3. Effect of TCDD on AhRR Expression in HaCaT Cells

To investigate the effect of TCDD on AhRR expression, we treated HaCaT cells with TCDD, and *AhRR* and *CYP1A1* expression were determined (Figure 3A). As a result, expression of *AhRR* and *CYP1A1* levels significantly increased following TCDD treatment. As shown in Appendix A, increases in AhRR and CYP1A1 mRNA expression were confirmed by TCDD in normal human epidermal keratinocytes (NHEK). Immunofluores-cence staining analysis showed that the expression of CYP1A1 was higher in TCDD dose-dependent HaCaT cells than in controls, being localized and enriched in the cytoplasm (Figure 3B). Simultaneously, *AhRR* levels increased following pAhRR (Figure 3C). This was significantly increased by pAhRR. These results suggest that TCDD induced *AhRR* expression in HaCaT cells similar to human psoriatic skin tissue (Figure 1B).

### 2.4. Effect of AhRR Overexpression on Pro-Inflammatory Cytokine Expression in HaCaT Cells

To further confirm the effect of AhRR on pro-inflammatory cytokine expression, we transfected HaCaT cells with control vector or pAhRR-HaCaT cells (pAhRR) to recapitulate the effect of hypomethylation of AhRR. This was a result of stably transfecting HaCaT cells with a control vector a high level of AhRR. Figure 4 shows that s100A1 and s100A12 (associated with epithelial cells); IL-6, IL-33, and IL-36 γ (keratinocyte inflammation cells); and IL-17A, IL-17C, and IL-22 (inflammatory cells secreted by T cells) mRNA expression levels were further increased in TCDD-treated pAhRR than in TCDD-treated normal HaCaT cells. These results suggested that pAhRR-HaCaT cells promote the expression of proinflammatory cytokines and eventually increase psoriasis-related factors in HaCaT cells.

### 2.5. Effect of AhRR Silencing on Pro-Inflammatory Cytokines in HaCaT Cells

To further confirm the role of TCDD in AhRR functioning, we transfected HaCaT cells with control siRNA or AhRR-siRNA. The results showed that AhRR-siRNA significantly inhibited the TCDD-induced expression of AhRR; CYP1A1; s100A1, s100A7, and s100A12 (associated with epithelial cells); IL-6 and IL-36γ (keratinocyte inflammation cells); and IL-17A, IL-17C, and IL-22 (inflammatory cells secreted by T cells) compared with control siRNA (Figure 5A). The results showed that AhRR-siRNA significantly inhibited the TCDD-induced CYP1A1 protein compared with control siRNA (Figure 5B). These results indicate that skin inflammation is involved with AhRR. Moreover, these findings suggested that when the AhRR gene is silenced, skin inflammation decreases.

### 2.6. Effect of TCDD on the AhRR and NF-κB Signaling Pathway in HaCaT Cells

To confirm whether TCDD is involved in the AhRR and NF-κB signaling pathway, we treated HaCaT cells with TCDD or TCDD with bay 11-7082 (NF-κB inhibitor). The results showed significantly decreased mRNA expression of AhRR and s100A1, s100A7, and s100A12 (associated with epithelial cells); IL-6 and IL-36γ (keratinocyte inflammation); and IL-17C and IL-22 (inflammatory cells secreted by T cells) in pre-treated bay 11-7082 (Figure 6). It was confirmed that AhRR increased by TCDD (AhR ligand) is also regulated in NF-κB transcriptional regulation.

To investigate the transcriptional activity of NF-κB, we transfected HaCaT cells with a plasmid containing the human NF-κB promoter region in a luciferase construct. Figure 6B shows that an increase in NF-κB promoter activity was detected in HaCaT cells treated with TCDD. In contrast, pretreatment with bay 11-7082 in HaCaT cells was decreased compared to cells treated only with TCDD. In addition, in identifying translocation of NF-κB, we found that TCDD markedly increased the translocation of NF-κB pp-50 from the cytoplasm to the nucleus (Figure 6C). These data suggest that TCDD induced pro-inflammatory cytokine expression via NF-κB pp50 in HaCaT cells.

To verify whether TCDD-AhRR is involved in the NF-κB signaling pathway, we treated pAhRR with TCDD or pretreated bay 11-7082 before TCDD, and pro-inflammatory cytokine concentrations were determined. pAhRR-TCDD-treated cells induced s100A1 and s100A12 (associated with epithelial cells); IL-6 and IL-36γ (keratinocyte inflammation); and IL-17C and IL-22 (inflammatory cells secreted by T cells) mRNA expression, which was significantly reduced by pretreatment with bay 11-7082 (Figure 6D). pAhRR-TCDD-treated cells induced s100A7 (associated with epithelial cells), IL-6 (keratinocyte inflammation), and IL-22 (inflammatory cell secreted by T cells) protein production, which was significantly reduced by pretreatment with bay 11-7082 (Figure 6E). Taken together, these results indicate that AhRR is involved in psoriatic skin inflammation induced by the NF-κB signaling pathway.

## 3. Discussion

In this study, we identified hypomethylation and overexpression of *AhRR* in the lesional epidermal skin of patients with psoriasis, and TCDD reacted with AhRR, AhR repressor, to confirm the effect of increasing pro-inflammatory cytokines in psoriasis skin tissues along with HaCaT cells. We also identified the underlying mechanism of action of AhRR on HaCaT cells. In conclusion, the AhR agonist TCDD increases pro-inflammatory cytokines associated with psoriasis skin tissue through activation of the AhRR/NF-κB signaling pathway.

In our previous study, TCDD was found to regulate the expression of AhR-related factors and cytokines in peripheral blood mononuclear cells (PBMCs) as well as CD4+ T cells from patients with psoriasis [9,23]. On the basis of this study, we investigated the increased expression level of AhRR in patients with psoriasis to understand the role of AhRR. The *AhRR* gene is significantly elevated in both the epidermis and dermis of the lesional skin of patients with psoriasis. This might suggest that the expression of the *AhRR* gene is related to inflammatory skin diseases. A recent study determined that common environmental factors such as cigarette smoke induce DNA hypomethylation of *AhRR* in subclinical atherosclerosis [24]. Therefore, further exposure to environmental causes can lead to changes in DNA methylation, resulting in diseases. In our study, hypomethylation of the *AhRR* gene was dominant in current smoking patients with psoriasis in the body loci of the gene. The relationship between cigarette smoke and *AhRR* hypomethylation is suggested to be closely related. Further studies are required to confirm the importance of positional differences in methylation and *AhRR* hypomethylation in cigarette smoke. In addition, a recent study also identified the effects of inflammatory cytokines from cigarette smoke [25]. On the basis of this study, we confirmed that the effects of cigarette smoke-induced pro-inflammatory cytokines in HaCaT cells. As TCDD is a component of cigarette smoke, the mRNA expression of pro-inflammatory cytokines significantly increased by TCDD via the AhR-related pathway [6,26].

Pro-inflammatory cytokines were expected to decrease in pAhRR-transfected HaCaT cells, but conversely increased significantly. AhRR enhances inflammatory response production by effector T cells in the inflamed gut [27]. Further, another investigation reported that AhRR mRNA expression increased due to TCDD treatment in AhRR tg mouse organ, but CYP1A1, a marker of AhR activation, was decreased more so than in normal mice [28]. According to these studies, pro-inflammatory cytokine regulation by AhRR is differently involved for each cell type. Therefore, AhR activity appears to be partially involved in feedback regulation through AhRR, depending on the immune cells of barrier organs, such as the skin [29].

CYP1A1 is one of the xenobiotic metabolizing enzymes (XMEs), one that is induced by polycyclic aromatic hydrocarbons (PAHs), with the most potent inducer of CYP1A1 being TCDD [30]. Chronic exposure to excessive toxic AhR ligands can lead to peroxidation and tissue destruction, owing to a high concentration of CYP enzymes [31]. Moreover, expression of the CYP1A1 gene is regulated by a heterodimeric transcription factor consisting of an aryl hydrocarbon receptor, a ligand-activated transcription factor [30]. A previous report demonstrated that S100 proteins constitute a family of calcium-binding proteins that play a major role in various cell types [32,33]. It is well known that S100 protein is a histological hallmark of PS skin lesions [33,34,35]. In addition, the increase in epidermal S100 protein in the skin tissue and blood of patients with psoriasis was found to be associated with disease severity via the NF-κB signaling pathway [36,37]. In chronic inflammatory skin diseases, an increase in pro-inflammatory cytokines (e.g., *IL-17*, *IL-22*, and *IL-6*) was confirmed, with these pro-inflammatory cytokines being known to be regulated by several signaling pathways, including RORγt and NF-κB [36,37].

Another finding surrounding the AhR–NF-κB mechanism shows that AhR interacts with NF-κB (RelA and RelB) [37]. We found that pro-inflammatory cytokine expression is downregulated by the NF-κB inhibitor bay 11-7082. A previous investigation demonstrated that AhR and AhRR are competitively combined with ARNT and enter the nucleus with NF-κB (RelB) [38]. It is composed of NF-κB heterodimer consisting of Rel and pp50 proteins [37]. As IκBα is phosphorylated, Rel and pp50 enter the nucleus and induce activation. Since BAY 11-7082 is an IκBα phosphorylation and NF-κB inhibitor, we confirmed pp50 activation.

We investigated the AhRR/NF-κB signaling pathway as such. TCDD-treated pAhRR; s100A1 and s100A12 (associated with epithelial cells); IL-6 and IL-36γ (keratinocyte inflammation); and IL-17C and IL-22 (inflammatory cells secreted by T cells) expression were reduced by pretreatment with bay 11-7082. Our findings showed that the intensity of the inflammatory response is modulated by the regulation of AhRR expression. Therefore, AhRR-induced inflammation of skin diseases are regulated through the NF-κB signaling pathway. AhRR induces CYP1A1 depletion, and, accordingly, relatively abundant AhR ligands might induce non-canonical AhR signaling pathway such as NF-kB activation. AhR has a canonical and non-canonical pathway, and there is an AhR pathway that controls gene expression regardless of TCDD treatment [39,40]. CYP1A1 deficiency in intestinal epithelial cells has been shown to increase the availability of AhR ligands [41]. AhRR acts as a repressor of AhR by interfering with AhR-ARNT binding in the canonical pathway [42]. However, in our study, AhRR did not counteract AhR in non-canonical pathways, but rather acted synergistically on the NF-κB signaling pathway. AhRR induces CYP1A1 depletion and accordingly relatively abundant AhR ligands might induce non-canonical AhR signaling pathway, such as NF-κB activation. It is assumed that demethylation of AhRR and consequent increase in AhRR expression probably occurs to compensate for these detrimental effects by other pathways. Meanwhile, increased AhRR protein induces the paradoxical expression of S100 proteins (associated with epithelial cells) or pro-inflammatory cytokines such as IL-17A, IL-17C, and IL-22 (inflammatory cells secreted by T cells) in TCDD-induced HaCaT cells.

## 4. Materials and Methods

### 4.1. Subjects

Patients with psoriasis and healthy volunteers (five males and five females) were recruited in the Department of Dermatology at Hallym University Kangnam Sacred Heart Hospital. Diagnosis of psoriasis was made by clinical and histological observation. Skin tissue samples were taken from the lesional skin from patients or normal areas from healthy controls. Peripheral blood mononuclear cells (PBMC) were taken from blood of psoriasis patients or from healthy controls. All subjects were asked about their age, job, habitat, dietary habits, and whether or not they smoked, and all gave their informed consent for the study (Hallym University Kangnam Sacred Heart Hospital). All methods and protocols were carried out in accordance with relevant guidelines and regulations by the Hallym University, Kangnam Sacred Heart Hospital, Human Research Ethics Committee of Hallym Institutional Review Board (Hallym 2012-05-45).

### 4.2. AhRR Methylation Analysis (Infinium HumanMethylation450 BeadChip)

Genomic DNA was isolated from the peripheral blood mononuclear cells (PBMC) of psoriasis patients and PBMC of healthy volunteers. Genomic DNA was isolated using a QIAamp^®^ DNA mini kit (Qiagen, Hilden, Germany) following the manufacturer’s instructions. Purified genomic DNA was processed for bisulfide conversion and subsequent methylation assays. DNA samples were delivered to Macrogen Inc., Korea. Genome-wide methylation profiles were generated using Illumina 450 K methylation arrays. Isolated DNA (100 ng) was employed to measure DNA methylation. All bioinformatics analyses were performed in Macrogen Inc., Korea.

### 4.3. Methylation-Specific PCR (MSP)

Genomic DNA was obtained from HaCaT cells using the QIAamp^®^ DNA Mini kit (Qiagen, Hilden, Germany). A total of 30 ng of DNA was submitted to methylation analysis performed using Episcope^®^ MSP kit (Takara Biotechnology Co., San Jose, CA, USA). PCR primers were designed using Meth Primer (http://www.urogene.org/methprimer/index1.html, (accessed on 1 April 2020), AhRR M left AGTATAGTGAGGATGGTGTTAGGTC, AhRR M right CTCGACGAATAAAAAATAAACGAA, AhRR UM left AGTATAGTGAGGATGGT GTTAGGTC, AhRR UM right ACTCAACAAATAAAAAATAAACAAA), and qPCR was performed using a LightCycler96 instrument (Roche, Basel, Switzerland).

### 4.4. Quantitative PCR

HaCaT cells (5 × 10^5^ cells/mL) were exposed to TCDD (500 nM) for 24 h. Total RNA was isolated per the manufacturer’s instructions using TRIzol (Invitrogen, Carlsbad, CA, USA). Two micrograms of RNA were reverse-transcribed using PrimeScriptTM RT Master Mix (Takara Biotechnology, CO., Ltd., Kusatsu, Japan) as per the manufacturer’s protocol. Synthesized cDNA was used in the RT-qPCR assay. Moreover, SYBR Green^TM^ Premix Ex TaqTM II Kit (Takara Biotechnology) was used for performing RT-qPCR, using the following primers: AhR (sense sequence 5′-CAAATCCTTCCAAGCGGCATA-3′, anti-sense sequence 5′-CGCTGAGCCTAAGAACTGAAAG-3′), GAPDH (sense sequence 5′-GTGGATATT GTTGCCATCA ATGACC-3′, anti-sense sequence 5′-GCCCCAGCCTTCTTCATG GTGGT-3′), AhRR (sense sequence 5′-GCGCCTCAGTGTCAGTTACC-3′, anti-sense sequence 5′-GAAGCCCAGATAGTCCACGAT-3′), CYP1A1 (sense sequence 5′-GATTGAGCACTGTCAGGAGAAGC-3′, anti-sense sequence 5′-ATGAGGCTCCAGGAGATAGCAG-3′), IL-6 (sense sequence 5′-AGACAGCCACTCACCTCTTCAG-3′, anti-sense sequence 5′-TTCTGCCAGTGCCTCTTTGCTG-3′), IL-8 (sense sequence 5′-GAGAGTGATTGAGAGTGGACCAC-3′, anti-sense sequence 5′-CACAACCCTCTGCACCCAGTTT-3′), IL-17A (sense sequence 5′-CGGACTGTGATGGTCAACCTGA-3′, anti-sense sequence 5′-GCACTTTGCCTCCCAGATCACA-3′), IL-17C (sense sequence 5′-CCC TGG AGA TAC CGT GTG GA-3′, anti-sense sequence 5′-GGG ACG TGG ATG AAC TCG G-3′), S100A1 (sense sequence 5′-GACCCTCATCAACGTGTTCCA-3′, anti-sense sequence 5′-CCACAAGCACCACATACTCCT-3′), S100A7 (sense sequence 5′-ACGTGA TGACAAGATTGACAAGC-3′, anti-sense sequence 5′-GCGAGGTAATTTGTGCCCTTT-3′), S100A12 (sense sequence 5′-AGCATCTGGAGGGAATTGTCA-3′, anti-sense sequence 5′-GCAATGGCTACCAGGGATATGAA-3′), IL-22 (sense sequence 5′-AGGCACTTACTGGCAACAGCA-3′, anti-sense sequence 5′-TGTCTGAGGTTTCACTGGTAAGG-3′), IL-25 (sense sequence 5′-CAGGTGGTTGCATTCTTGGC-3′, anti-sense sequence 5′-GAGCCGGTTCAAGTCTCTGT-3′), IL-33 (sense sequence 5′-CAGGTGGTTGCATTCTTGGC-3′, anti-sense sequence 5′-GAGCCGGTTCAAGTCTCTGT-3′), IL-36r primer synobiological (Sino Biological US Inc.). Measurements were conducted in triplicate, and the fold change values were calculated using the ΔCt method. qPCR was performed using LightCycler96 instrument (Roche, Switzerland). Results were obtained from at least three independent experiments.

### 4.5. HaCaT Cell Culture Conditions

HaCaT cells from Hallym University Kangnam Sacred Heart Hospital were obtained in February 2020. HaCaT cells were cultured in Dulbecco’s modified Eagle’s medium (DMEM) containing 10% (*v*/*v*) heat-inactivated fetal bovine serum (FBS; Invitrogen, Carlsbad, CA, USA). A total of 1% (*v*/*v*) 1 × 10^4^ units/mL penicillin was employed this context. The cells were grown in a humidified atmosphere at 37 °C and 5 % CO_2_. Thereafter, HaCaT cells (5 × 10^5^ cells/mL) were exposed to TCDD (500 nM) or bay 11-7082 (3 nM) (Sigma-Aldrich, St. Louis, MO, USA).

### 4.6. RNA Interference

Control siRNAs and AhRR (50 nM) were synthesized by Santa Cruz Biotechnology (Dallas, TX, USA). HaCaT cells (5 × 10^5^ cells/mL) were transiently transfected with siRNA using Lipofectamine^TM^ RNAiMax (Invitrogen, Carlsbad, CA, USA) as per the manufacturer’s instructions.

### 4.7. Western Blot

HaCaT cells were lysed using the RIPA Lysis solution (Thermo Scientific, Waltham, MA, USA) and centrifuged for 30 min at 10,000× *g.* HPDF lysates were subjected to sodium dodecyl sulfate polyacrylamide gel electrophoresis and transferred onto PVDF membranes (Millipore Inc., Billerica, MA, USA). Membranes were blocked with 5% skim milk solution and incubated overnight at 4 °C with the following primary antibodies: AhRR (1:1000) and CYP1A1 (1:1000) (Abcam, Cambridge, MA, USA), and GAPDH (1:1000) (Santa Cruz Biotechnology). After incubation, the membranes were washed in Tris-buffered saline–0.1% Tween-20 buffer and treated with peroxidase-conjugated anti-rabbit or anti-mouse immunoglobulin G (IgG). The blots were visualized with HRP-conjugated secondary antibodies and ECL system (ATTO Corporation, Tokyo, Japan).

### 4.8. AhRR Overexpression

The pLNPH Bglll vector was used for the insertion of AhRR. Its implementation was established by Cosmogenetech Inc. (Seoul, Korea). The final concentrations for the AhRR plasmids were 0.75 μg/mL. The concentration for lentivirus transduction was 8 × 10^5^ transducing units of lentivirus antibiotic resistance constructed using ampicillin (100 μg/mL). Transfections were performed using Lipofectamine 2000 Reagent (Invitrogen, Carlsbad, CA, USA) in HaCaT cells (8 × 10^5^ cells/mL).

### 4.9. Immunohistochemical Staining

Human skin tissue sections were deparaffinized in xylene and dehydrated using graded alcohols. After dehydration, sections were placed in sodium citrate buffer (10 mM, pH 6.0), brought to pressure in a microwave, and held at pressure for 2 min. After a brief rinse, each section was incubated with the primary anti-AhRR antibody (Abcam, Cambridge, MA, USA) for 24 h at room temperature, followed by a biotinylated secondary antibody at 1:200 for 1 h. Finally, for nuclear counterstaining, Vectashield mounting medium was used along with DAPI. All skin sections were captured and visualized using a microscope (Leica Microsystems, Wetzlar, Germany).

### 4.10. Immunocytochemical Staining

HaCaT cells were fixed with 4% paraformaldehyde for 30 min. Cells were permeabilized with 0.2% Triton X-100 in 1% bovine serum albumin (BSA) for 10 min, blocked with 3% BSA for one hour at room temperature, and incubated overnight at 4 °C with anti-AhRR, anti-CYP1A1 (Abcam, Cambridge, MA, USA), NF-kB pp50, and p50 (Santa Cruz Biotechnology). Cells were then incubated in goat anti-mouse HRP, goat anti-Rb HRP (Millipore), and the second antibody conjugated with FITC (Abcam) at 1:200 for 1 h. Stained HaCaT cells were captured and visualized using a microscope (Leica Microsystems, Wetzlar, Germany). For nuclear counterstaining, Vectashield mounting medium was used.

### 4.11. Luciferase Reporter Assay

A luciferase reporter assay was performed using a Dual Luciferase Reporter Assay Kit (BPS Bioscience, San Diego, CA, USA), according to the manufacturer’s protocol. The luciferase reporter construct was co-transfected with NF-κB reporter into HaCaT cells using Lipofectamine 2000 (Invitrogen, Carlsbad, CA, USA), according to the manufacturer’s guidelines. TCDD was administered after 24 h. To obtain the normalized luciferase activity for the NF-κB reporter, we subtracted the background luminescence and calculated the ratio of firefly luminescence from the NF-κB reporter to Renilla luminescence from the control Renilla luciferase vector. The relative luciferase activity was measured using an automatic plate reader (BioTek Instruments, Seoul, Korea).

### 4.12. Enzyme-Linked Immunosorbent Assay (ELISA)

IL-6, IL-22, and S100A7 quantitation by enzyme-linked immunosorbent assay (ELISA) protein expression levels in HPDFs after treatment with TCDD and pAhRR were evaluated by ELISA. HaCaT cells were exposed to various concentrations of TCDD and pAhRR for 24 h, and the supernatant from the cultured cells was collected. IL-6, IL-22, and S100A7 levels were measured using ELISA (Biolegend, San Diego, CA, USA). Briefly, each well was blocked with blocking buffer for 2 h and washed with buffer. Antibodies against IL-6, IL-22, and S100A7 were added to the media and incubated for 1 h. A substrate solution and stop solution were introduced sequentially, and the optical density of each well was determined within 30 min using an automatic plate reader (BioTek Instruments Korea Ltd.).

### 4.13. Statistical Analysis

Results are shown as the mean ± standard error of the mean. Statistical significance of differences between groups was assessed by one-way analysis of variance (ANOVA) for factorial comparisons and by Tukey’s multiple comparison tests for multiple comparisons using Prism for Windows software (GraphPad 5 Software Inc., San Diego, CA, USA). Results were obtained from at least three independent experiments.

## 5. Conclusions

In conclusion, we confirmed AhRR hypomethylation and increased AhRR gene expression in lesion skin from psoriasis patients, and increased expression of psoriasis-associated proinflammatory cytokines and S100 protein via AhRR-NF-kB signaling. This study investigated the mechanism by which psoriasis is related to smoking. This suggests that controlling the hypomethylation or hyperactivation of AhRR can suppress the exacerbation of these chronic skin diseases by environmental factors, such as smoking. Thus, this study provides evidence for the inhibitory effect of AhRR, specifically AhRR silencing, on critical pro-inflammatory cytokines in the case of skin diseases.

## 6. Patents

We hold a patent for overexpression of aryl hydrocarbon receptor inhibitor (AhRR) in skin keratinocytes (application no. 10-2020-0132481).

## Figures and Tables

**Figure 1 ijms-22-12715-f001:**
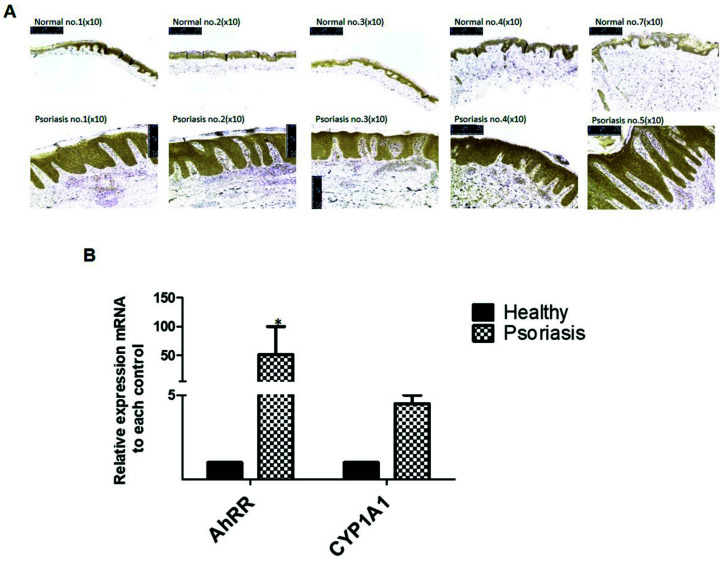
Expression of AhRR in healthy human skin and lesional skin of psoriasis. (**A**) Immunohistochemical staining was performed for AhRR on healthy and psoriasis epithelial skin samples. Positive cells appeared brown (for all objective X10/*n* = 5 groups). (**B**) Relative expression of AhRR and CYP1A1 target genes was confirmed in healthy and psoriasis skin samples. The control group indicated healthy people, and the test group indicated psoriasis patients. Values are means ± SEM. Statistical significance of differences between groups was assessed by one-way analysis of variance (ANOVA) for factorial comparisons and by Tukey’s multiple comparison tests for multiple comparisons. All experiments of at least five donors each per condition are shown. * *p* < 0.05 vs. healthy control.

**Figure 2 ijms-22-12715-f002:**
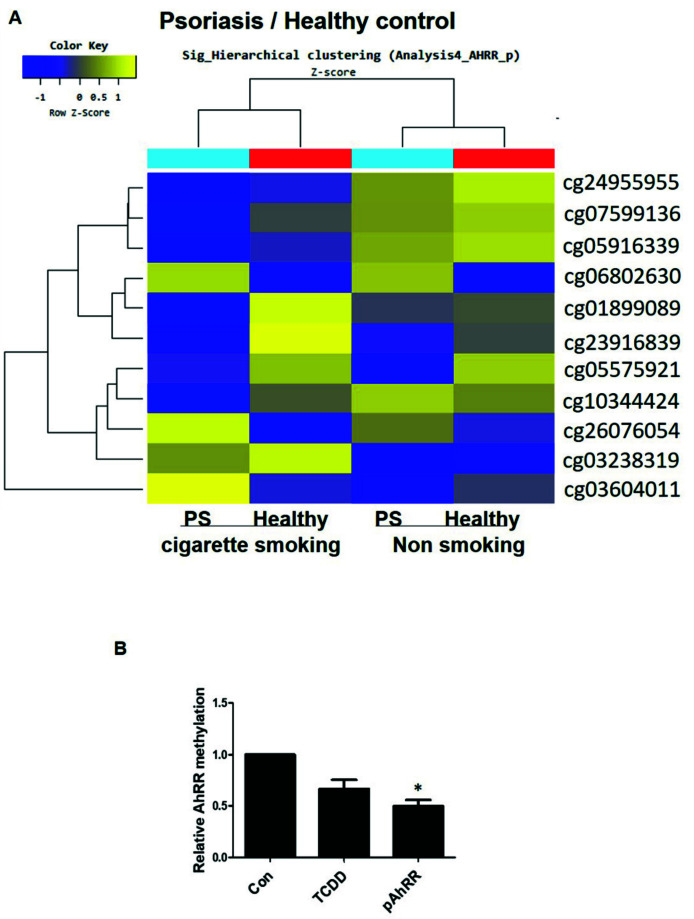
Effect of hypomethylation in PBMC of psoriasis and in HaCaT cells. (**A**) Data corresponding to the *AhRR* gene of healthy and psoriasis PBMC samples are displayed as a color-coded gene-by-sample heatmap, with rows (AhRR genes) and columns (control and psoriasis patients or smoking and non-smoker groups) sorted by hierarchical clustering. Dark blue is hypomethyl and light blue is less hypomethylation. Dark yellow is hypermethylation and light yellow is less hypermethylation. (**B**) HaCaT cells were transfected with pAhRR (0.3 μg) for 24 h and treated with TCDD (500 nM) for 24 h. AhRR methylation was examined by methylation-specific PCR (MSP). Values are means ± SEM. All experiments were performed in at least three independent experiments. * *p* < 0.05 vs. control.

**Figure 3 ijms-22-12715-f003:**
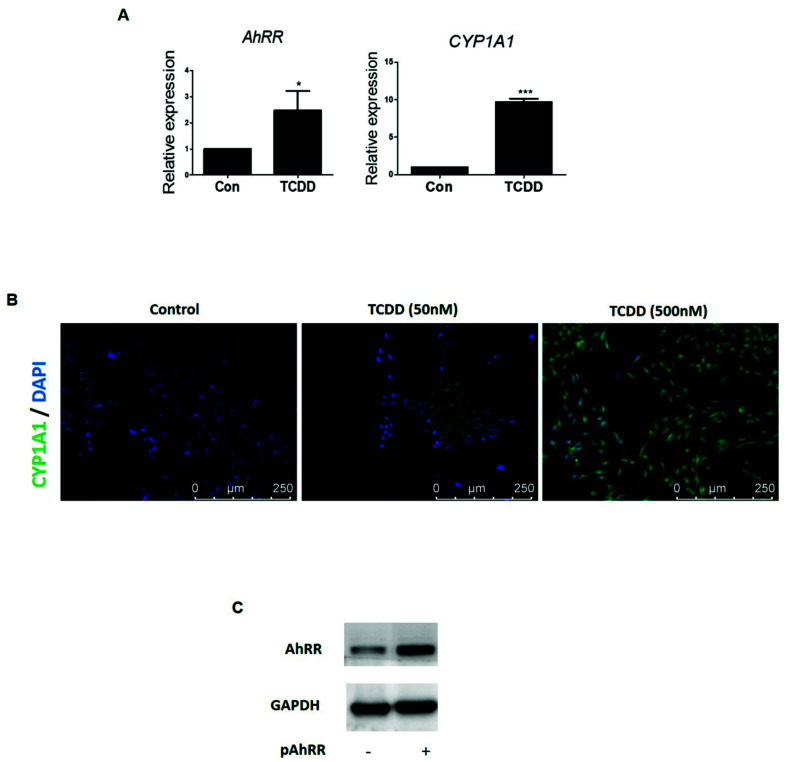
Effect of TCDD on AhRR expression in HaCaT cells. HaCaT cells were treated with TCDD 500 nM for 24 h. (**A**) *AhRR* and *CYP1A1* mRNA expression was examined by qPCR. (**B**) Localization of CYP1A1 was determined by immunofluorescence staining (scale bar: 250 μm). (**C**) HaCaT cells were treated with TCDD 500 nM (24 h) and transfected with pAhRR for 48 h. *AhRR* protein level was examined by Western blot. Values are means ± SEM. All experiments of at least three independent experiments. * *p* < 0.05 vs. control, *** *p* < 0.001 vs. control.

**Figure 4 ijms-22-12715-f004:**
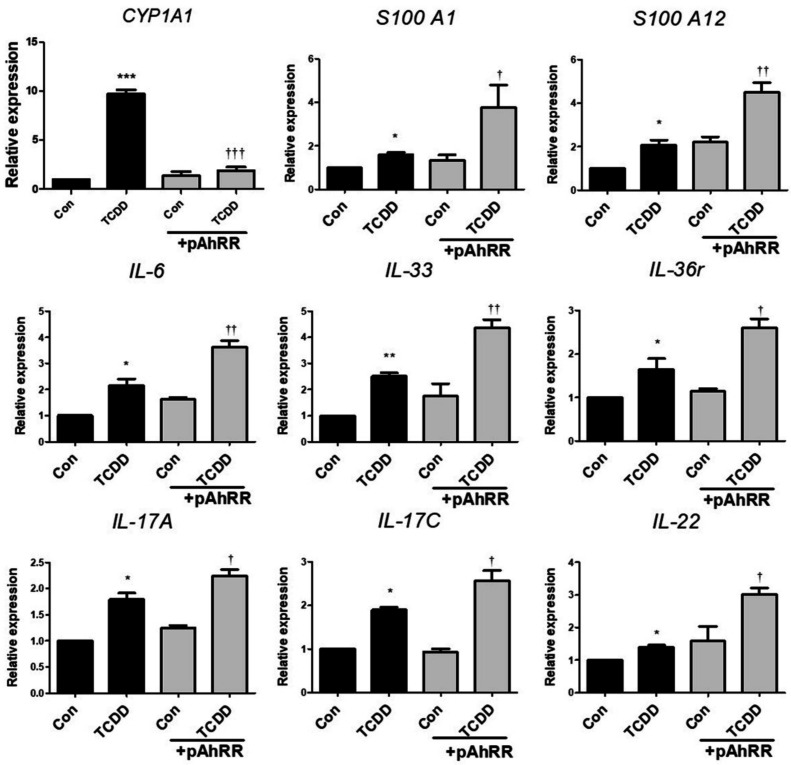
Effect of AhRR overexpression on pro-inflammatory cytokine expression in HaCaT cells. Pro-inflammatory cytokine expression mRNA expression in HaCaT cells was examined by qPCR. Values are means ± SEM. Statistical significance of differences between groups was assessed by one-way analysis of variance (ANOVA) for factorial comparisons and by Tukey’s multiple comparison tests for multiple comparisons. All experiments were performed in at least three independent experiments. * *p* < 0.05 vs. control, ** *p* < 0.01 vs. control, *** *p* < 0.001 vs. control, † < 0.05 vs. TCDD, †† < 0.01 vs. TCDD, ††† < 0.001 vs. TCDD.

**Figure 5 ijms-22-12715-f005:**
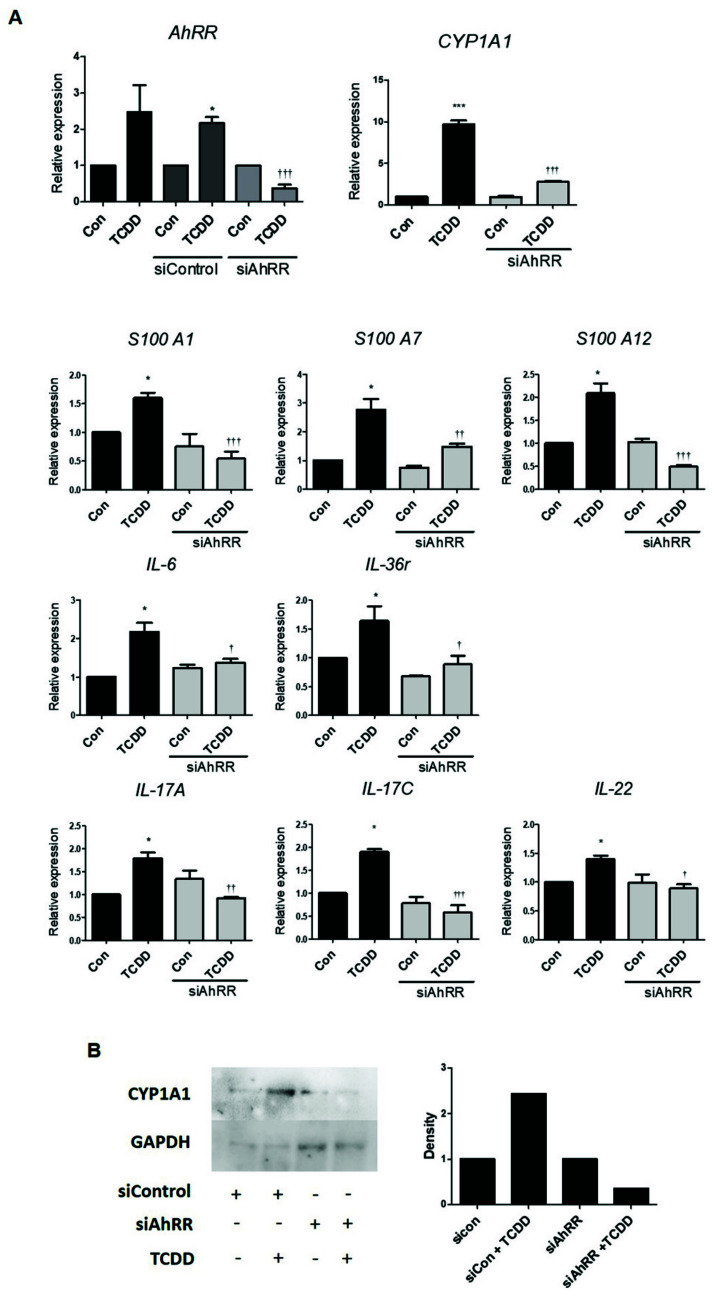
Effect of AhRR silencing on pro-inflammatory cytokines in HaCaT cells. HaCaT cells were transfected with siControl (10 nM) and siAhRR (50 nM) for 4 h and treated with TCDD (500 nM) for 24 h, and mRNA expression levels were examined by qPCR (**A**) and Western blot (**B**) Values are means ± SEM. All experiments were performed in at least three independent experiments. * *p* < 0.05 vs. control, *** *p* < 0.001 vs. control, † *p* < 0.05 vs. TCDD, †† *p* < 0.01 vs. TCDD, ††† *p* < 0.001 vs. TCDD.

**Figure 6 ijms-22-12715-f006:**
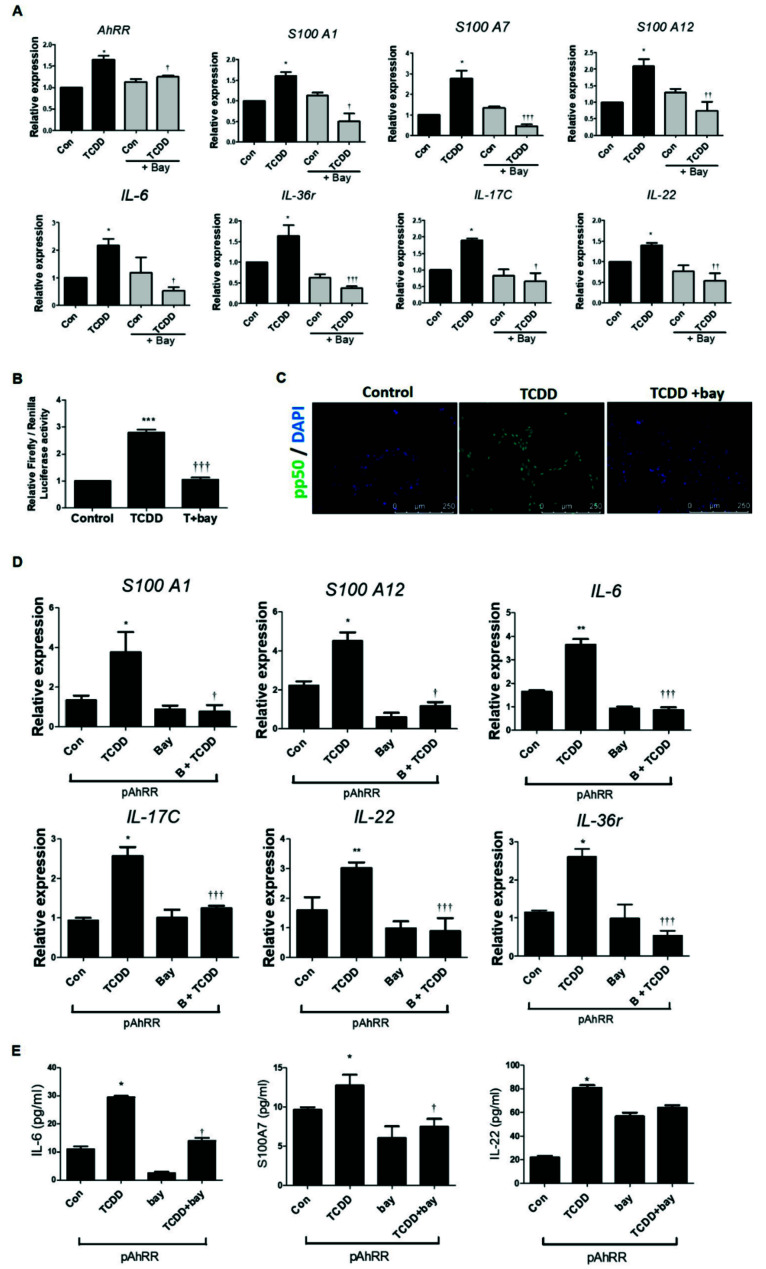
Effect of TCDD on AhRR by NF-κB signaling pathway in HaCaT cells. HaCaT cells were pre-treated with bay 11-7082 (NF-κB inhibitor) at 3 nM for 1 h, and then treated with TCDD (500 nM) for 24 h. **(A)** mRNA expression levels of pro-inflammatory cytokines in HaCaT cells along with TCDD were examined by qPCR. **(****B)** The NF-κB promoter region of activation was determined by the luciferase reporter assay. **(****C)** Translocation of NF-κB pp50 was determined by immunofluorescence staining (scale bar: 250μm). (**D**) mRNA expression levels of pro-inflammatory cytokines in HaCaT cells along with AhRR-overexpressed HaCaT cells were examined by qPCR. (**E**) Protein production of pro-inflammatory cytokines in HaCaT cells along with AhRR-overexpressed HaCaT cells were examined by ELISA. Values are means ± SEM. * *p* < 0.05 vs. control, ** *p* < 0.01 vs. control, *** *p* < 0.001 vs. control, † *p* < 0.05 vs. TCDD, †† *p* < 0.01 vs. TCDD, ††† *p* < 0.001 vs. TCDD, ** *p* < 0.01 vs. pAhRR+TCDD, ††† *p* < 0.001 vs. pAhRR+TCDD.

## Data Availability

Not applicable.

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
