# Peer review of "Aryl Hydrocarbon Receptor Repressor Is Hypomethylated in Psoriasis and Promotes Psoriasis-like Inflammation in HaCaT Cells"

_ijms, 2021, doi:10.3390/ijms222312715_

Round 1
Reviewer 1 Report
This manuscript by Um and Chung et al., entitled ‘Aryl Hydrocarbon Receptor Repressor Is Hypomethylated in 2 Psoriasis and Promotes Psoriasis-like inflammation in HaCaT 3 Cells' investigates the transcriptional expression of aryl hydrocarbon receptor repressor (AhRR) in lesional skin of Psosoriasis patients. They found AhRR gene up-regulation in these samples, associated with hypomethylation of the gene. Furthermore, the authors demonstrate that over-expression of AhRR enhance a proinflammatory response via NF-kB signal pathway.
Overall, the manuscript is of general interest and add a new piece in the understanding the AhR/CYP1A1 axis in psoriasis disease.
However, the authors should address specific concerns before publication.
- In line 77, authors should specify the reason why they analysed CYP1A1 mRNA levels in psoriasis and health samples. This is stated only in discussion. This might help the reader.
- In Material and Methods sections, authors should insert paragraph regarding RNA extraction/cDNA synthesis and methods used for transcriptional analysis. Moreover, they should also add a table with qPCR primer sequences.
- Although the authors state in the line 77 that they found a significant CYP1A1 transcriptional up-regulation in sample-derived psoriasis patients (Figure 1), this comparison lacks statistical analysis. Please, insert statistical analysis.
- Fig 2A: It is not clear what rows represent. Figure 2A should represents methylation pattern of AhRRgene (rows) between different samples (column). Please, specified what each raw represent in the figure or in the legend, and which gene loci have been analyzed . Moreover, authors should insert in Method section information about bioinformatic analysis.
- The authors should insert Methylation-specific primer sequences in Method section
- Although authors state in line 95 ‘ AhRR overexpression (pAhRR) and TCDD-treated HaCaT cells showed significant hypomethylation of the AhRR gene than non-treated cells. ‘, statistical analysis is not showed for TCDD-treated cells. Please insert it.
- Figure 5: Authors should include in the manuscript a control experiment in which they demonstrate an effective AhRR transcriptional silencing after siRNA strategy in HaCaT cells.
- Figure 6: Figure panel 6A lacks reference letter ‘A’
- In figure 4 authors should discuss the data comparison between HeCat-con cells and HeCaT-con +pAhRR cells, and add statistical analysis in figure. It seems that over-expression of AhRR in HeCaT genes might have an impact of S100A12, IL6 and IL33 gene expression.
- Figure 3 legend: Two typo errors in line 120 and 121. Furthermore, legend lacks complete statistical references about p-value
- In Line 100-102 typo errors makes sentences confused
- Line 131, ‘ vector or a control vector’…. typo error?
- The authors should include figure 6 in the paragraph 2.6, where it is cited for the first time
Author Response
This manuscript by Um and Chung et al., entitled ‘Aryl Hydrocarbon Receptor Repressor Is Hypomethylated in 2 Psoriasis and Promotes Psoriasis-like inflammation in HaCaT 3 Cells' investigates the transcriptional expression of aryl hydrocarbon receptor repressor (AhRR) in lesional skin of Psosoriasis patients. They found AhRR gene up-regulation in these samples, associated with hypomethylation of the gene. Furthermore, the authors demonstrate that over-expression of AhRR enhance a proinflammatory response via NF-kB signal pathway.
Overall, the manuscript is of general interest and add a new piece in the understanding the AhR/CYP1A1 axis in psoriasis disease.
However, the authors should address specific concerns before publication.
- In line 77, authors should specify the reason why they analysed CYP1A1 mRNA levels in psoriasis and health samples. This is stated only in discussion. This might help the reader. Several environmental carcinogens such as TCDD are known to affect CYP1A1 expression [32]. Therefore, it was confirmed as a marker whether AhR was activated by TCDD.
- In Material and Methods sections, authors should insert paragraph regarding RNA extraction/cDNA synthesis and methods used for transcriptional analysis. Moreover, they should also add a table with qPCR primer sequences. adds
- Although the authors state in the line 77 that they found a significant CYP1A1 transcriptional up-regulation in sample-derived psoriasis patients (Figure 1), this comparison lacks statistical analysis. Please, insert statistical analysis. Statistical significance of differences between groups was assessed by one-way analysis of variance (ANOVA) for factorial comparisons and by Tukey’s multiple comparison tests for multiple comparisons.
- Fig 2A: It is not clear what rows represent. Figure 2A should represents methylation pattern of AhRRgene (rows) between different samples (column). Please, specified what each raw represent in the figure or in the legend, and which gene loci have been analyzed. rows (AhRR genes) and columns (control and psoriasis patients or smoking and non-smoker groups). In our study, hypomethylation of the AhRR gene is dominant in current smoking patients with psoriasis in the body loci of the gene. Moreover, authors should insert in Method section information about bioinformatic analysis All bioinformatics analyses were performed in Macrogen Inc., Korea
- The authors should insert Methylation-specific primer sequences in Method section
- Although authors state in line 95 ‘ AhRR overexpression (pAhRR) and TCDD-treated HaCaT cells showed significant hypomethylation of the AhRR gene than non-treated cells. ‘, statistical analysis is not showed for TCDD-treated cells. Please insert it. Statistical significance of differences between groups was assessed by one-way analysis of variance (ANOVA) for factorial comparisons and by Tukey’s multiple comparison tests for multiple comparisons
- Figure 5: Authors should include in the manuscript a control experiment in which they demonstrate an effective AhRR transcriptional silencing after siRNA strategy in HaCaT cells. data was added.
- Figure 6: Figure panel 6A lacks reference letter ‘A’ A attached
- In figure 4 authors should discuss the data comparison between HeCat-con cells and HeCaT-con +pAhRRcells, and add statistical analysis in figure. It seems that over-expression of AhRR in HeCaT genes might have an impact of S100A12, IL6 and IL33 gene expression. inserted in figure legend.
Thank you so much for the good point. We think that over-expression of AhRR is definitely effective for psoriasis markers.
- Figure 3 legend: Two typo errors in line 120 and 121. Furthermore, legend lacks complete statistical references about p-value Modified to TCDD * p < 0.05 vs. control,
- In Line 100-102 typo errors makes sentences confused
Modified to These results showed that AhRR hypomethylation was induced in both TCDD-treated HaCaT cells and AhRR overexpressing HaCaT cells.
- Line 131, ‘ vector or a control vector’…. typo error? Modified to control vector
- The authors should include figure 6 in the paragraph 2.6, where it is cited for the first time Figure 6 has been added.

Reviewer 2 Report
Um et al. have contributed an interesting and novel study. In brief, the authors attempt to link environmental pollutants (TCDD herein) with expression and hypomethylation of the Repressor of AchR activity (AhRR) and the skin disease, psoriasis. They show that AhRR is elevated in the skin of psoriatic patients and use appropriate cell line models to manipulate AhRR levels and function to measure typical psoriatic manifestations such as increased cytokine levels and NFkB activity.
There are numerous technical flaws which diminish my enthusiasm for the manuscript.
If these are overcome then I’d encourage a resubmission which I would gladly review for a second time.
Major comments
- Figure 1 : What is the purpose of CYP1A1 here? The rationale for its use is not clear in the text. The elevation of AhRR in psoriatic skin is compelling. Two more images from the n=5 groups should be shown.
- Figure 2B : A western blot to show pAhRR expression should be shown. Given that the AhRR antibody (figure 4A) is of low quality, the AhRR cDNA may be tagged (myc or HA) to show exogenous expression. The manuscript could also be more greatly strengthened if the AhRR cDNA is mutated to remove putative methylation sites. This essential control should be included in this experiment and others and would further support the role of hypomethylation in the claimed biological processes.
- Figure 3. A western blot and/or immunofluorescence to show TCDD-mediated elevation of AhRR and CyP1A1 expression should be shown. A time course or TCDD dose response would be useful.
- Figure 4A. The western blot for AhRR is unconvincing and indicates numerous non-specific bands. Regardless, a 2-fold overexpression is poor and unlikely to have the biological effects subsequently claimed.
- Figure 4B. The IF images are clearly not 100x as claimed (more like 10x) and are out of focus.
- Figure 4C. Cytokine measurement by qPCR is unacceptable. The authors should measure secreted cytokines in the supernatant of HACAT cells by ELISA (note : we prefer mesoscale).
- Figure 5: AhRR knockdown by siRNA should be confirmed by western blot. What is the transfection efficiency of the siRNA in these cells?
- Figure 6C: The images shown are of much too low magnification to make any conclusions. Plus, the claimed 400x magnification is not accurate. 4x is more realistic.
Minor comments
- Figure 1A : The scale bar doesn’t align with the image. What does the scale represent?

Author Response
Comments and Suggestions for Authors
Um et al. have contributed an interesting and novel study. In brief, the authors attempt to link environmental pollutants (TCDD herein) with expression and hypomethylation of the Repressor of AchR activity (AhRR) and the skin disease, psoriasis. They show that AhRR is elevated in the skin of psoriatic patients and use appropriate cell line models to manipulate AhRR levels and function to measure typical psoriatic manifestations such as increased cytokine levels and NFkB activity.
There are numerous technical flaws which diminish my enthusiasm for the manuscript.
If these are overcome then I’d encourage a resubmission which I would gladly review for a second time.
Major comments
- Figure 1: What is the purpose of CYP1A1 here? Several environmental carcinogens such as TCDD are known to affect CYP1A1 expression. Therefore, it was confirmed as a marker whether AhR was activated by TCDD. The rationale for its use is not clear in the text. The elevation of AhRR in psoriatic skin is compelling. Two more images from the n=5 groups should be shown. 5 All images were attached.
- Figure 2B: A western blot to show pAhRR expression should be shown. Given that the AhRR antibody (figure 4A) is of low quality, the AhRR cDNA may be tagged (myc or HA) to show exogenous expression. The manuscript could also be more greatly strengthened if the AhRR cDNA is mutated to remove putative methylation sites. This essential control should be included in this experiment and others and would further support the role of hypomethylation in the claimed biological processes. As shown in Figure 2C.
- Figure 3. A western blot and/or immunofluorescence to show TCDD-mediated elevation of AhRR and CyP1A1 expression should be shown. A time course or TCDD dose response would be useful. As shown in Figure 3B.
- Figure 4A. The western blot for AhRR is unconvincing and indicates numerous non-specific bands. Regardless, a 2-fold overexpression is poor and unlikely to have the biological effects subsequently claimed. As shown in Figure 2C.
- Figure 4B. The IF images are clearly not 100x as claimed (more like 10x) and are out of focus. Modified
- Figure 4C. Cytokine measurement by qPCR is unacceptable. The authors should measure secreted cytokines in the supernatant of HACAT cells by ELISA (note: we prefer mesoscale). Cytokines were identified. As shown in Figure 6E.
- Figure 5: AhRR knockdown by siRNA should be confirmed by western blot. What is the transfection efficiency of the siRNA in these cells? As shown in Figure 5B. When confirmed by the qPCR result, there is a transfection efficiency of 85% or more.
Figure 6C: The images shown are of much too low magnification to make any conclusions. Plus, the claimed 400x magnification is not accurate. 4x is more realistic. data was corrected.
Minor comments
- Figure 1A : The scale bar doesn’t align with the image. What does the scale represent? A scale bar is displayed.

Round 2
Reviewer 1 Report
Accept in present form
Author Response
Thanks for the review. Good luck with your research.
Reviewer 2 Report
While my support for this manuscript remains, there are some unexpected inconsistencies in the work that have diminished my enthusiasm :-
1- Figure 2C. The text claims that " ..Simultaneously, AhRR levels increased in following TCDD treatment and pAhRR. (Fig. 2C."
But this is not correct; the indicated figure suggests that the levels of AhRR protein appear identical between control and TCDD treatment in HACAT cells.
Like figure 2C, figure 3 also looks at the role of TCDD on AhRR expression in HACAT cells. Firstly, these two similar experiments would be better combined in a single figure. Secondly, the results given in 3A are inconsistent with 2C. In the latter, TCDD increases AhRR expression.
Please clarify these differences.
Figure 4A: The figure legend states that HACAT cells were transfected for 24hrs yet line 146 claims "This is a result of stably transfecting HaCaT cells with a control vector a high level of AhRR."
Firstly, it is hard to believe that 24hr expression of a protein results in 100% uniform expression in the cytoplasm of every cell, which is shown. This image looks like background immunofluorescence with different exposure times. Is is stable or transient (24hr) expression.
I'd recommend removal of this figure. A western blot to indicate overexpression will make a suitable replacement.
Minor
1- Methods : Line 364 The authors claim that "HaCaT cells were tested and mycobacterial contamination was verified". If this true then all HACAT experiments are invalid. I am sure that this is a typographical error.
2- a few typographical errors are apparent in this revised version.
Author Response
1- Figure 2C. The text claims that " .Simultaneously, AhRR levels increased in following TCDD treatment and pAhRR. (Fig. 2C."
But this is not correct; the indicated figure suggests that the levels of AhRR protein appear identical between control and TCDD treatment in HACAT cells.
Like figure 2C, figure 3 also looks at the role of TCDD on AhRR expression in HACAT cells. Firstly, these two similar experiments would be better combined in a single figure. Secondly, the results given in 3A are inconsistent with 2C. In the latter, TCDD increases AhRR expression.
In fact, the increase time by TCDD is 24h, and the pAhRR is a total of 48 hours after plasmid transfection, so it is correct to remove TCDD because the time is different. I couldn't cut it because I had to show it as a full blot. Therefore, Figure 2C was modified and added as Figure 3C.
Please clarify these differences.
Figure 4A: The figure legend states that HACAT cells were transfected for 24hrs yet line 146 claims "This is a result of stably transfecting HaCaT cells with a control vector a high level of AhRR."
Firstly, it is hard to believe that 24hr expression of a protein results in 100% uniform expression in the cytoplasm of every cell, which is shown. This image looks like background immunofluorescence with different exposure times. Is is stable or transient (24hr) expression.
I'd recommend removal of this figure. A western blot to indicate overexpression will make a suitable replacement.
Figure 4A was removed, Protein increase by pAhRR is added to Figure 3C.
Minor
1- Methods : Line 364 The authors claim that "HaCaT cells were tested and mycobacterial contamination was verified". If this true then all HACAT experiments are invalid. I am sure that this is a typographical error. This sentence is incorrect. Removed..
2- a few typographical errors are apparent in this revised version. Corrected.

Round 3
Reviewer 2 Report
I have no further comments and congratulations on a fine piece of work.
Author Response
Thank you for the reboot. I wish you good results in your research as well.